
# Mott transition in a cavity-boson system:
# A quantitative comparison between theory and experiment

Rui Lin[1], Christoph Georges[2], Jens Klinder[2], Paolo Molignini[3], Miriam Büttner[4],
Axel U. J. Lode[4], R. Chitra[1], Andreas Hemmerich[2,5] and Hans Keßler[2*]

**1** Institute for Theoretical Physics, ETH Zürich, 8093 Zurich, Switzerland
**2** Zentrum für Optische Quantentechnologien and Institut für Laser-Physik,
Universität Hamburg, 22761 Hamburg, Germany
**3** Cavendish Laboratory, 19 JJ Thomson Avenue, Cambridge CB3 0HE, United Kingdom
**4** Institute of Physics, Albert-Ludwig University of Freiburg,
Hermann-Herder-Straße 3, 79104 Freiburg, Germany
**5** The Hamburg Center for Ultrafast Imaging,
Luruper Chaussee 149, 22761 Hamburg, Germany

* hkessler@physnet.uni-hamburg.de

## Abstract

The competition between short-range and cavity-mediated infinite-range interactions in a cavity-boson system leads to the existence of a superfluid phase and a Mott-insulator phase within the self-organized regime. In this work, we quantitatively compare the steady-state phase boundaries of this transition measured in experiments and simulated using the Multiconfigurational Time-Dependent Hartree Method for Indistinguishable Particles. To make the problem computationally feasible, we represent the full system by the exact many-body wave function of a two-dimensional four-well potential. We argue that the validity of this representation comes from the nature of both the cavity-atomic system and the Bose-Hubbard physics. Additionally, we show that the chosen representation only induces small systematic errors, and that the experimentally measured and theoretically predicted phase boundaries agree reasonably well. We thus demonstrate a new approach for the quantitative numerical modeling for the physics of the superfluid–Mott-insulator phase boundary.

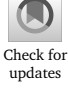
# 1    Introduction

During the past decade, experimental and theoretical progress using quantum gases to realize models of solid state physics has made it possible to study many-body effects in isolated and highly controllable scenarios [1–3]. In particular, the interplay between light and matter creates a unique platform for the exploration of a plethora of exotic behaviors in quantum systems [4–15]. Compared to traditional solid state systems, light-matter systems have a simpler nature, because they comprise much fewer particles and have easily tunable system parameters. These advantages enable the study of a wide range of toy models, and the achieved knowledge can be further applied to systems with complex band structures and interactions in solid state physics and material science.

Many-body effects in ultracold atomic systems have seen an enduring interest, particularly the coherence between particles in the superfluid phase and its loss in the Mott-insulator phase of a lattice system. The transition between these two phases is driven by the competition of the tunneling processes and the on-site interactions, and was first realized by controlling an optical lattice potential in cold-atom systems in three [16] and two dimensions [17,18], respectively. This transition can also be realized in a system as sketched in Fig. 1, where the optical potential is self-organized due to the coupling of atoms to an optical cavity. Driven by an external laser and with cavity-mediated interactions, effectively two-dimensional Bose-Einstein condensates (BECs) can self-organize into a lattice [7]. As the drive intensity increases, a transition between a self-organized superfluid (SSF) phase and a self-organized Mott-insulator (SMI) phase has been observed experimentally [15, 19] and investigated theoretically [20–24]. This transition

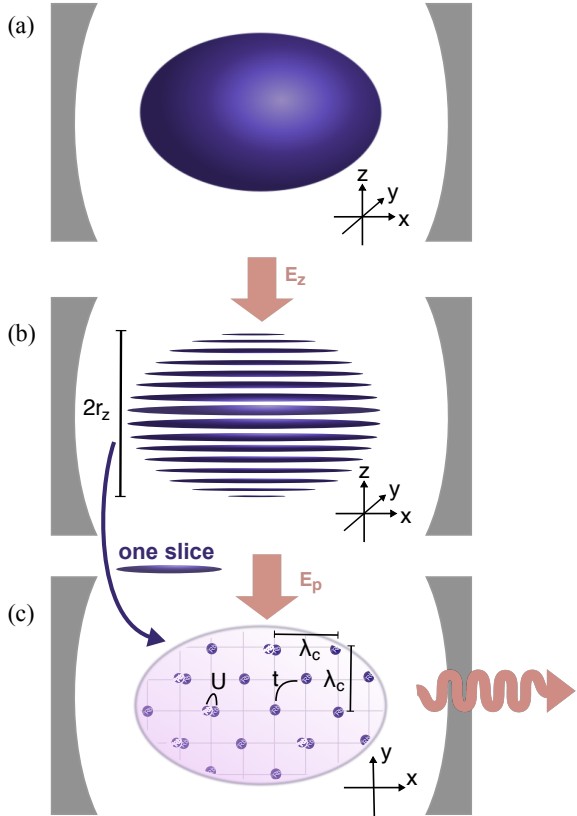

Figure 1: Sketch of the experimental system. The atoms are first prepared as (a) a three-dimensional BEC, and then cut into (b) two-dimensional slices by an external laser pump along $z$ direction. (c) Due to the pumping laser along $y$ direction and the interplay with the cavity, they finally self-organize into a checkerboard lattice with wavelength $\lambda_c$ along both $x$ and $y$ directions. The onset of the self-organization can be detected by the intracavity photons. After the checkerboard lattice is formed, the system can be mapped to a Bose-Hubbard model with tunneling strength $t$ and on-site interaction $U$.

stems from a combination of the short-range interaction due to $s$-wave scattering between the atoms and the infinite-range interaction mediated by the cavity [25, 26]. The cavity-BEC system thus realizes a quantum-optical version of the Bose-Hubbard model, where any pair of lattice sites is coupled by global infinite-range interactions [19, 27, 28].

Hitherto, a direct quantitative comparison between experiment and theory regarding to the SSF–SMI transition has not been presented because of the enormous computational effort required. However, this comparison is crucial for future applications like machine learning techniques, which have recently been applied to various physical systems [29–33], including ultracold atomic systems [34–36]. Because of their exact control of system parameters and their shorter time scales in data collection, quantitative numerical simulations provide complementary access to data for the training of neural networks for experimental systems.

In this work, we perform quantitative numerical simulations of the SSF–SMI phase transition, in particular of the phase diagram, by employing the Multiconfigurational Time-Dependent Hartree Method for Indistinguishable Particles (MCTDH-X) [37–44], and validate the simulated results with the experimental ones. MCTDH-X captures many-body effects beyond the Gross-Pitaevskii mean-field limit, including but not limited to the coherence between the atoms. In contrast, treating the current cavity-BEC system in the mean-field limit can capture

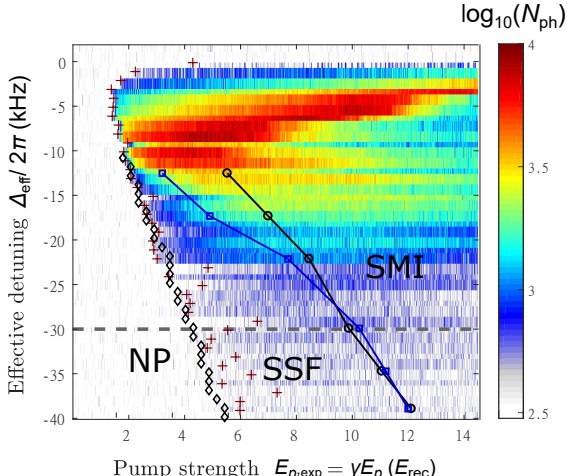

Figure 2: The steady-state phase diagram identifying the normal BEC phase (NP), the self-organized superfluid phase (SSF) and the self-organized Mott-insulator phase (SMI). It is plotted against effective cavity-pump detuning $\Delta_{\text{eff}}$ and pump strength $E_{p,\text{exp}} = \gamma E_p$ in units of the recoil energy $E_{\text{rec}}$, where $\gamma = 1.36$ is a calibration factor between the pump strength used in experiments and simulations. To determine the experimental NP–SSF boundary (dark red crosses), we use the slow ramping protocol with ramping time $T_r = 40$ ms and measure the intracavity photon number (background color). The boundary is then defined by the rapid increase in the photon number. It is compared to the simulated NP–SSF boundary (black diamonds). To determine the experimental SSF–SMI boundary (black circles), we use the fast ramping protocol with $T_r = 20$ ms and measure the momentum space density. The boundary is then defined by the rapid increase in the central peak width. It is compared to the simulated SSF–SMI boundary (blue squares) which is obtained through our proposed simplification scheme. The simplification scheme induces systematic errors in the predicted boundary of roughly $\pm 0.5 E_{\text{rec}}$. The black and blue lines are guides to the eyes, and the gray dashed line marks the detuning $\Delta_{\text{eff}}$ used in Figs. 5 and 6.

the self-organization, but fails to describe the Mott insulation. In order to keep the computational complexity within a tractable range, we construct a simplification scheme for the simulations by exploiting the nature of the cavity-BEC system and the superfluid–Mott-insulator transition. This simplification scheme nevertheless retains the many-body essence of the system to a satisfactory degree, and quantitatively reproduces the phase boundary in agreement with the experiments in a wide parameter range. The comparison is summarized in the phase diagram in Fig. 2.

This work is organized as follows. In Sec. 2, we introduce the system as well as the experimental setup, and we describe our experimental protocol for obtaining the experimental phase diagram. In Sec. 3, we first briefly introduce MCTDH-X, and then propose a simplification scheme, which is well adapted and specialized to the cavity-BEC system and MCTDH-X. In Sec. 4, we compare the experimental and simulation results, and discuss the origin of the discrepancy between them. Moreover, we compare and cross validate our MCTDH-X scheme against existing approach based on Wannier functions and the Bose-Hubbard model. Finally, we draw conclusions in Sec. 5.

## 2 Experimental setup and measurement protocol

### 2.1 Cavity-BEC system and superfluid–Mott-insulator transition

The experimental system, as sketched in Fig. 1, consists of a laser-driven Bose-Einstein condensate (BEC) of $N_{3D} = 5.5 \times 10^4$ rubidium-87 ($^{87}$Rb) atoms dispersively coupled to a high-finesse optical cavity with strength $\Omega_g$. The atoms are magnetically trapped in a three-dimensional harmonic potential with trapping frequencies $(\omega_x, \omega_y, \omega_z) = 2\pi \times (25.2, 202.2, 215.6)$ Hz. In the absence of an external drive, the ensemble forms a Thomas-Fermi cloud with measured radii $(r_x, r_y, r_z) = (26.8, 3.3, 3.1)\,\mu$m [Fig. 1(a)]. The three-dimensional atomic cloud is overlapped with the fundamental mode of the high-finesse optical cavity oriented along the $x$ direction. The cavity resonance frequency $\omega_c$ and wave vector $k_c$ correspond to a wavelength of $\lambda_c = 803$ nm and a recoil energy of $E_{rec} = \hbar^2 k_c^2 / 2m_{Rb} = \hbar \times 2\pi \times 3.55$ kHz. The cavity has a field decay rate of $\kappa = 2\pi \times 4.45$ kHz comparable to the recoil frequency, and therefore operates in the sub-recoil regime [45]. After the initial preparation, the atomic cloud is then loaded into an external optical lattice oriented along the $z$ direction, which is given by $E_{ext}(z) = E_z \cos^2(2\pi z/\lambda_z)$ with wavelength $\lambda_z = 803$ nm and depth $E_z = 12.5\,E_{rec}$. The strong external lattice suppresses tunneling along the $z$ direction and reduces the system into effective two-dimensional slices spanned on the $x$-$y$ plane, as illustrated in Fig. 1(b).

After preparing and loading the BEC into $E_{ext}(z)$, the ensemble is transversely pumped along the $y$ direction by a laser with effective pump strength $E_p$ and frequency $\omega_p = 2\pi\lambda_p/c$, which forms an effective standing-wave optical lattice $E_y(y) = E_p \cos^2(2\pi y/\lambda_p)$. We work in the dispersive regime $\omega_p \ll \omega_a$ using pump light at a wavelength of $\lambda_p = 803$ nm. This is far detuned from the relevant atomic transition of $^{87}$Rb at $\lambda_a = 795$ nm. We note that the atoms and the cavity are both red-detuned from the pump light $\Delta_a = \omega_p - \omega_a < 0$, $\Delta_c = \omega_p - \omega_c < 0$.

Combining all the aforementioned components of the setup, we can write down the full many-body Hamiltonian of the cavity-BEC system [7, 46, 47],

$$\hat{\mathcal{H}} = \int dx\,dy\,dz\,\hat{\Psi}^\dagger\left(\frac{\mathbf{p}^2}{2m_{Rb}} + V_{trap} + V_{opt}\right)\hat{\Psi} + \frac{g_{3D}}{2}\int dx\,dy\,dz\,\hat{\Psi}^\dagger\hat{\Psi}^\dagger\hat{\Psi}\hat{\Psi}, \tag{1a}$$

$$V_{trap} = \frac{m_{Rb}}{2}(\omega_x^2 x^2 + \omega_y^2 y^2 + \omega_z^2 z^2), \tag{1b}$$

$$\begin{aligned} V_{opt} = &-E_p \cos^2(k_c y) - E_z \cos^2(k_c z) \\ &+ \hbar U_0 |\alpha|^2 \cos^2(k_c x) + \sqrt{\hbar E_p |U_0|}(\alpha + \alpha^*)\cos(k_c x)\cos(k_c y). \end{aligned} \tag{1c}$$

Here, $\hat{\Psi} \equiv \hat{\Psi}(x, y, z)$ is the atomic annihilation operator, $m_{Rb} = 1.44 \times 10^{-25}$ kg is the mass of the $^{87}$Rb atoms, $g_{3D}$ is the atom-atom contact interaction strength, and $U_0 = \Omega_g^2/\Delta_a = -2\pi \times 0.36$ Hz is the single-atom light shift. The cavity field, pumping laser, and external lattice are near resonance $\lambda_c \approx \lambda_p \approx \lambda_z$, and for clarity we denote the wavelengths and wave vectors along all three directions as $\lambda_c$ and $k_c$ respectively in Eq. (1) and in the rest of this work. A summary of the experimental parameters is given in Appendix A.

The cavity field is treated as coherent light and represented by its expectation value $\alpha = \langle a \rangle$, where $a$ is the annihilation operator of the cavity field. The expectation value $\alpha$ follows the equations of motion [7, 46, 47]

$$\partial_t \alpha = [i(\Delta_c - N_{3D} U_0 B) - \kappa]\alpha - i\sqrt{\frac{E_p |U_0|}{\hbar}} N_{3D}\theta, \tag{2a}$$

$$\theta = \int dx\,dy\,dz\,\rho(x, y, z)\cos(k_c x)\cos(k_c y), \tag{2b}$$

$$B = \int dx\,dy\,dz\,\rho(x, y, z)\cos^2(k_c x), \tag{2c}$$

where $\rho(x, y, z) = \langle \hat{\Psi}^\dagger \hat{\Psi} \rangle / N_{3D}$ is the spatial density distribution. Under this treatment, the cavity field effectively imposes a one-body potential upon the atoms, as evident in the second line of Eq. (1c). This treatment of the cavity field is legitimate as long as the cavity fluctuations $\langle \delta a^2 \rangle = \langle a^\dagger a \rangle - |\alpha|^2$ are small, which is indeed the case except near the self-organization boundary [23, 48–50], which will be introduced in detail below.

The atomic many-body wave function of the steady state of the cavity-BEC system can be obtained by solving Eqs. (1) and (2) self-consistently. While for small pump strengths the system remains in the normal BEC phase (NP), for pump strengths above a critical threshold, the atoms reduce the potential energy by self-organizing into a checkerboard lattice with lattice spacing $\lambda_c$ along the $x$ and $y$ directions as depicted in Fig. 1(c), and constructively scatter photons from the pump into the cavity [7–9, 47–49]. In a steady state, the dominant part $[\cos(k_c x) \cos(k_c y)]$ of the cavity-induced potential has an effective depth

$$E_{cb} = \left| \frac{2 E_p U_0 N_{3D} \theta (\Delta_c - N_{3D} U_0 B)}{(\Delta_c - N_{3D} U_0 B)^2 + \kappa^2} \right|. \tag{3}$$

This self-organization transition can be mapped to the Hepp-Lieb normal–superradiant phase transition of the Dicke model [51–54], and is accompanied by the spontaneous breaking of the $\mathbb{Z}_2$ symmetry, which is reflected by the sign of $\theta$. A positive (negative) $\theta$ corresponds to an even (odd) lattice configuration. In our experimental system this symmetry is well established, and the system spontaneously breaks into either configuration upon self-organization [55].

Deep in the self-organized phase, the atoms progressively localize on the checkerboard lattice sites as the pump strength increases and the induced optical potential deepens. Coherence between atoms at different lattice sites gradually decays, leading to a second transition from the SSF phase to the SMI phase [15, 23, 24]. During this transition, cavity fluctuations are indeed minimal [23, 48–50], validating our mean-field treatment of the cavity field. The SSF and SMI phases behave similar to the superfluid and Mott-insulator phases, respectively, of the usual Bose-Hubbard model

$$H_{BH} = -t \sum_{\langle i,j \rangle} \left( b_i^\dagger b_j + b_j^\dagger b_i \right) + \frac{U}{2} \sum_i b_i^\dagger b_i^\dagger b_i b_i, \tag{4}$$

where $b_i$ is the annihilation operator for bosonic atoms at the $i$-th lattice site, $t$ is the tunneling strength, $U$ is the on-site interaction, and $\langle i, j \rangle$ indicates the summation is over nearest neighbors. In this model, a superfluid is characterized by a fluctuating particle number per site and phase coherence of the whole ensemble due to large tunneling between different lattice sites. On the contrary, in a Mott-insulator, phase coherence is lost, the particle fluctuations vanish and the number of atoms per lattice site is fixed due to suppressed tunneling. The differences between the two phases lead to distinct behaviors in various quantities, including the variance of the on-site atom number $\text{Var} = \langle (b_i^\dagger b_i)^2 \rangle - \langle b_i^\dagger b_i \rangle^2$ [56–58] and the momentum space density distribution [15–18, 22–24, 59–63]

$$\tilde{\rho}(\mathbf{k}) = \langle \hat{\Psi}^\dagger(\mathbf{k}) \hat{\Psi}(\mathbf{k}) \rangle. \tag{5}$$

Since the former quantity is hard to measure experimentally [56], we choose $\tilde{\rho}(\mathbf{k})$ as our main quantity of interest for defining the phase boundary. As the system enters the Mott-insulator phase from the superfluid phase, a significant increase in the full width at half maximum (FWHM) $\mathcal{W}$ of the central peak in the momentum space density distribution can be observed [15–18, 23, 24, 59–61] accompanying the loss of phase coherence. The transition between the two phases is thus smooth and has only weak criticality. For a $d$-dimensional system, it is in the same universality class as a $(d+1)$-dimensional $XY$ model [64].

In the cavity-BEC system, the total number of atoms enters the equation of motion Eq. (2) and effectively modifies the cavity detuning. Meanwhile, the number of atoms per site, equivalent to the filling factor in the Bose-Hubbard model, is an important ingredient in determining the SSF–SMI boundary [17, 59, 65–67]. Therefore, a quantitative comparison between experiment and theory necessitates an estimate of the number of atoms in each two-dimensional slice as well as at each lattice site near the center of the harmonic trap. For simplicity, we assume a uniform distribution of the atoms in the central cuboid of the three-dimensional harmonic trap, such that $x/r_x$, $y/r_y$ and $z/r_z$ are all within the interval $[-1/2, 1/2]$. In this region, since the two-dimensional slices are $\lambda_c/2$ apart from each other along the $z$ direction, there are $2r_z/\lambda_c \approx 8$ slices in total and each slice contains roughly $N_{2D} \approx 6,900$ atoms. Once the system enters the self-organized phase, the atoms in each slice will further form a lattice with two lattice sites per area of $\lambda_c^2$. There are thus $2r_x r_y/\lambda_c^2 = 275$ lattice sites in the considered rectangle on each slice, and each of the lattices contains $\nu \approx 25$ atoms.

## 2.2 Measurement protocol

The comparison between the experimental and simulated phase diagrams involves both the NP–SSF and the SSF–SMI boundaries for the steady state. In experiments, we fix the effective detunings

$$\Delta_{\text{eff}} = \Delta_c - \frac{1}{2}N_{3D}U_0 \,, \tag{6}$$

while ramping up the pump strength linearly from zero to $E_{p,\text{exp}} = 14.5E_{\text{rec}}$ within a time $T_r$. There is a trade-off, which will be described in detail below, when choosing an appropriate ramping time, and we choose two different ramping times for the measurements of different observables to best approximate the steady-state phase boundaries.

In the vicinity of the NP–SSF boundary, the photonic behavior is dominating due to significant cavity fluctuations. As the cavity decay rate is small in comparison to the effective detuning $\kappa < |\Delta_{\text{eff}}|$, the cavity field experiences a retardation effect when crossing the steady-state NP–SSF boundary [9]. As a result, the dynamical NP–SSF boundary shifts towards higher pump strength for shorter ramping time $T_r$, and converges to the steady-state boundary with long $T_r$. With a ramping time of $T_r = 40$ ms, the hysteresis area is negligibly small and the steady-state boundary can be well approximated [9].

On the contrary, deep inside the self-organized phase, the cavity fluctuations vanish and atomic behavior becomes dominant, rendering particularly atom loss a key factor. A decrease in the atom number effectively increases $|\Delta_{\text{eff}}|$ when both $\Delta_c$ and $U_0$ are negative [cf. Eq. (6)], and it generally indicates that a higher pump strength $E_p$ is required to achieve the same lattice depth [cf. Eq. (3)]. Therefore, all phase boundaries are shifted towards higher pump strengths when atom loss occurs. Since a longer ramping time implies a larger atom loss and hence a larger shift in the boundary, a fast ramp with $T_r = 20$ ms is thus preferred for the measurement of the steady-state SSF–SMI boundary.

After understanding the dynamical effects on the boundaries, we first use the slow ramp $T_r = 40$ ms for the determination of the NP–SSF boundary. During the ramp, we record the transmitted photons leaking through one of the cavity mirrors using a single photon counting module (SPCM), and scale them with the detection efficiency to obtain the intracavity photon number $N_{\text{ph}}$. This is plotted in logarithmic scale in Fig. 2, where a background count offset originating from diffuse light is subtracted from the measured signal. We can then determine the phase boundary according to the threshold of the corrected photon number $N_{\text{ph}} \approx 300$. The measured NP–SSF boundary will later be compared to the simulated one as a calibration. In Fig. 2, the effects of the atom loss can already be observed for more negative detunings at the onset of the self-organization, where the measured NP–SSF boundary is slightly shifted to

higher pump strengths. Similar effects can also be observed for the SSF–SMI boundary, which we will discuss in detail in Sec. 4.1.

We then use the fast ramp $T_r = 20$ ms to determine the SSF–SMI boundary, which is extracted from the momentum space density distribution $\tilde{\rho}(\mathbf{k})$ [15, 16, 18, 21, 23]. To measure the momentum distribution we repeat the experiment several times where we stop at a certain pump strength for ballistic expansion of the sample. After switching off all the trapping potentials and a 25 ms long time of flight, we detect the momentum distribution using single-shot absorption images. Thereafter, we extract the width of the central peak from the distribution, and mark the SSF–SMI boundary at the pump strength where the width starts to increase. The measured SSF–SMI boundary is marked as the black line with circles in the phase diagram Fig. 2. With the fast ramping protocol $T_r = 20$ ms, the measured dynamical NP–SSF is indeed significantly shifted towards larger pump strengths when compared to the slow ramping protocol (see Appendix B). Nevertheless, the cavity field and the induced potential converge to the steady-state values soon after the system dynamically enters the self-organized phase, as verified in Ref. [9]. This significantly reduces the retardation effect on the dynamical SSF–SMI boundary.

Caution needs to be taken when analyzing the experimental measurements. The experimentally calibrated pump strength $E_{p,\mathrm{exp}}$ is different from the pump strength $E_p$ entering the Hamiltonian, because the experimental pump laser is not strictly monochromatic. The pump strength is calibrated by measuring the energy difference between the first and the third Bloch bands at zero quasi-momentum. This is done by an active modulation of the lattice depth and measuring the resonance frequency for parametric heating of the BEC [68, 69]. Such measurement considers effects from electromagnetic waves of all frequencies. The central peak is almost in resonance with the cavity frequency $\omega_c$, and has a linewidth of roughly 100 Hz. The linewidth of the pump is well within the cavity linewidth, which is equivalent to the cavity dissipation rate $\kappa$. Therefore, the central peak can fully contribute to the scattering of the cavity field. However, there are also two servo bumps with a frequency shift roughly of $\pm 2$ MHz from the cavity frequency $\omega_c$, which is large compared to the cavity linewidth. As a result, the light from the side peaks cannot scatter into the cavity or contribute to the effective cavity-induced potential of the atoms, and the effective pump strength $E_p$ entering Eqs. (1) and (2) is different from the experimental one $E_{p,\mathrm{exp}}$ obtained directly through calibration in experiments. These two pump strengths are related through a calibration factor

$$\gamma \equiv E_{p,\mathrm{exp}}/E_p > 1 \,, \tag{7}$$

which is not measurable through experiments without applying significant hardware changes to the system. To determine the factor $\gamma$, we need to compare the experimental and simulated phase diagrams, and require that they coincide with each other. This comparison will be performed in Sec. 3.2 after obtaining the simulated NP–SSF boundary.

# 3 Simulation methodology

## 3.1 Numerical method

We use the approach Multiconfigurational Time-Dependent Hartree Method for Indistinguishable Particles (MCTDH-X) to simulate the steady state of the system and extract the observables of interest [37–43], like the momentum space density distribution and the cavity field expectation value. MCTDH-X is able to solve problems beyond the Gross-Pitaevskii mean-field limit, and capture the correlations between atoms as well as quantum fluctuations in the many-body states. The method relies on a variational ansatz for the many-body state, which

is the permanent of multiple time-dependent optimized functions, or orbitals. The number of orbitals $M$ controls the simulation accuracy. Ideally, the exact solution of the numerical problem is found when an infinite number of orbitals is used [39, 40, 70]. MCTDH-X has been successfully applied for investigating the static and dynamic behaviors of Bose-Hubbard systems [22, 23, 62, 63, 71–74]. A more detailed description of the method can be found in Appendix C.

The number of orbitals used in a simulation depends on the nature of the quantum state of interest. For example, the formation of the cavity-induced potential and thereby the self-organization of the atoms can well be observed in the mean-field limit with $M = 1$ orbital [23, 38, 75]. In contrast, to correctly describe a Mott-insulator state, the number of orbitals should be at least as large as the number of lattice sites [22, 23, 43, 70, 76]. Since the required computational resources scale as $\binom{N-M+1}{M}$ [39], given the currently available computational softwares and hardwares, it is computationally unfeasible to simulate the full experimental cavity-BEC system with MCTDH-X. Therefore, we need to simplify the problem and reduce the number of orbitals and particles needed for the MCTDH-X simulations. We will now elaborate on the methodology for choosing this simplification.

## 3.2 Reduction of system dimensionality and rescaling of the contact interaction strength

The computational complexity can be significantly reduced by lowering the system dimensionality. We argue that the system can be well represented by a two-dimensional model, and determine the effective atom-atom interaction strength in this model.

In experiments, the system is divided into two-dimensional slices by the deep external optical lattice. The hopping between two slices is strongly suppressed by the strong external lattice, and thus the slices are independent from each other on the atomic level [cf. Eq. (1)]. On the other hand, atoms from all slices collectively contribute to the cavity field, and therefore they are strongly coupled to each other through the cavity [cf. Eq. (2)]. In order to represent the full system by one two-dimensional slice, we propose to decouple the slices by simulating Eqs. (1) and (2) with $N_{2D}$ atoms at $z = 0$, and using the scaling of parameters [47]

$$U_0 \mapsto \tilde{U}_0 = U_0 \frac{N_{3D}}{N_{2D}}, \quad \alpha \mapsto \alpha \sqrt{\frac{N_{2D}}{N_{3D}}}. \tag{8}$$

Under this scaling, the equations of motion for the cavity field [Eq. (2)] as well as the cavity-induced potential $V_{opt}$ [Eq. (1)] remain invariant for a fixed atomic density profile. We thus expect that the atomic many-body wave function of the two-dimensional system obtained from Eq. (8) approximately reproduces the wave function of the original system at $z = 0$,

$$|\Psi_{2D}(x, y)\rangle \approx |\Psi_{3D}(x, y, z = 0)\rangle. \tag{9}$$

This approach requires a knowledge of the effective contact interaction in the two-dimensional slice, which is crucial for the formation of Mott insulation. The strength $g_{2D}$ is estimated according to the harmonic trapping frequencies and the corresponding Thomas-Fermi radii [77],

$$N_{2D} g_{2D} = \frac{\pi m_{Rb}}{4} \frac{r_x^4 \omega_x^3}{\omega_y}, \tag{10}$$

which yields $g_{2D} \approx 0.34 \hbar^2 / m_{Rb}$ for $N_{2D} = 6,900$, as explained in detail in Appendix D.

With the effective single slice, we can simulate the physics of the realistic experimental system using MCTDH-X at different pump lattice depths $E_p$ and effective detunings $\Delta_{eff}$ in simulations, and as the first observable we choose the cavity field strength. The self-organization

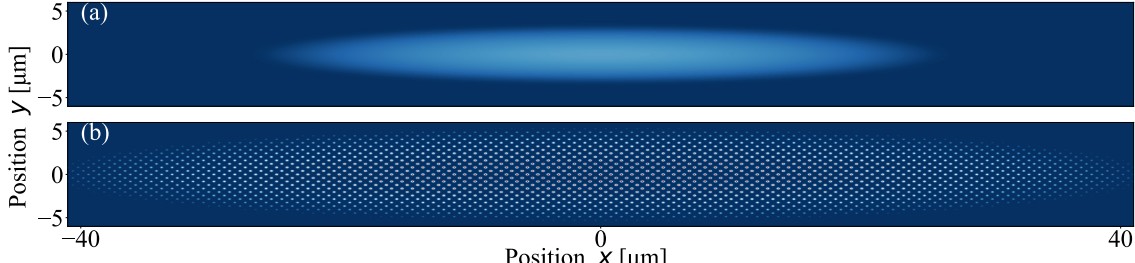

Figure 3: Real space density distributions $\rho(x)$ of (a) a normal BEC state and (b) a self-organized state of the full two-dimensional system simulated in the mean-field limit using MCTDH-X. The zooms of the two states around the center of the harmonic trap are shown in Figs. 5(s) and 5(t), respectively.

and the accompanying macroscopic activation of the cavity field can already be captured with sufficient precision by using $M = 1$ orbital in the mean-field limit. Exemplary real space density distributions of a normal BEC state with a Thomas-Fermi profile and a self-organized state with a checkerboard lattice are shown in Fig. 3. Along with the density distributions, these mean-field simulations also yield the cavity field expectation values $\alpha_{\mathrm{MF}}(E_p, \Delta_{\mathrm{eff}})$. The NP–SSF boundary can then be drawn at $|\alpha_{\mathrm{MF}}|^2 \approx 0.1$. Although this choice of threshold is different from the experimental one, it does not lead to a substantial difference in the predicted boundary due to the rapid increase of photon number across the boundary. Both criteria are chosen based on the analytically expected boundary and the respective limitations in experiments and simulations.

With the simulation results, the calibration factor $\gamma$ [cf. Eq. (7)] for the experimental pump strength can now be calculated to be $\gamma = 1.36$. This is determined by requiring that the measured NP–SSF boundary and the simulated one, which are fitted as $\Delta_{\mathrm{eff}}/2\pi = (-8.536 E_{p,\mathrm{exp}}/E_{\mathrm{rec}} + 6.305)$ kHz and $\Delta_{\mathrm{eff}}/2\pi = (-11.616 E_p/E_{\mathrm{rec}} + 5.834)$ kHz respectively, have the same slope as functions of pump strengths. For the fitting of the experimental boundary, only the data measured between cavity detunings $-2\pi \times 5$ kHz and $-2\pi \times 20$ kHz is taken into account, because atom loss already slightly shifts the boundary at more negative detunings. The experimental and simulated NP–SSF boundaries indeed collapse upon each other when this calibration factor is taken into account (cf. Fig. 2). The effective contact interaction strength $g_{\mathrm{2D}}$ and the calibration factor $\gamma$ are the last two system parameters to be determined for the comparison to the experimental system.

### 3.3 Four-well model

We now proceed to simulate the SSF–SMI transition. A proper description of this transition requires at least one orbital for each lattice site. Given the large number of atoms and lattice sites, it is impractical to simulate the quantum state of the full two-dimensional system, and thus a further simplification of the model is needed. Since the SSF–SMI transition is mainly driven by the competition between on-site interaction and hopping between nearest-neighboring sites, the loss of superfluidity of the whole system should already be quantitatively captured by a local representation. A minimal choice for such a local representation is a unit cell consisting of four lattice sites in the center of the harmonic trap, which is shown in Fig. 4(b).

This four-well potential can be described by the Hamiltonian

$$\hat{\mathcal{H}}_{4\mathrm{well}} = \int dx\, dy\, \hat{\Psi}^\dagger \left( \frac{\mathbf{p}^2}{2m_{\mathrm{Rb}}} + V_{4\mathrm{well}} \right) \hat{\Psi} + \frac{g_{\mathrm{2D}}}{2} \int dx\, dy\, \hat{\Psi}^\dagger \hat{\Psi}^\dagger \hat{\Psi} \hat{\Psi}, \tag{11}$$

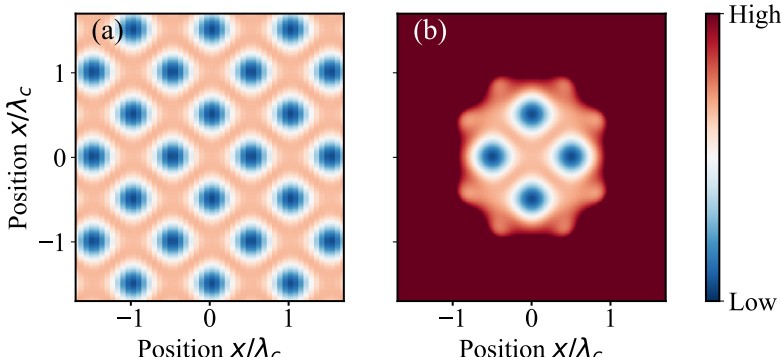

Figure 4: A comparison between (a) the cavity-induced lattice potential $V_{\text{opt}}(z = 0)$ [Eq. (1)] and (b) the four-well potential $V_{\text{4well}}$ [Eq. (12)]. Each of the four wells faithfully reproduces the lattice sites of the lattice.

where a tight non-harmonic confining potential is applied on top of the optical lattice

$$V_{\text{4well}} = \tilde{V}_{\text{opt}} + V_{\text{conf}}. \tag{12a}$$

The ideal confining potential should be relatively flat in the center of the system $x^2 + y^2 < \lambda_c^2$, but form a rapidly increasing wall surrounding the four wells at $x^2 + y^2 > \lambda_c^2$. This can be achieved by using an algebraic function of $x^2 + y^2$ with high power. For example, the following confining potential is chosen for our simulation:

$$V_{\text{conf}}(x, y) = 13 E_{\text{rec}}(x^2 + y^2)^4 / \lambda_c^8. \tag{12b}$$

We note that this is not the unique choice for the confining potential, and the simulated SSF–SMI boundary should not be sensitive to the choice (see Appendix E).

However, a straightforward implementation of the tight confining potential can easily distort the underlying optical lattice, because the equations of motion [Eqs. (1), (2)] are solved self-consistently and the solution can be very sensitive to slight changes of parameters, especially near the self-organization boundary. We thus make use of the previously simulated expectation value of the cavity field $\alpha_{\text{MF}}(E_p, \Delta_{\text{eff}})$ to determine the depths of the cavity-induced potential, i.e., $U_0|\alpha_{\text{MF}}|^2$ and $2\sqrt{\hbar E_p |U_0|}\text{Re}(\alpha_{\text{MF}})$, which is equivalent to using

$$\tilde{V}_{\text{opt}}(x, y) = V_{\text{opt}}(x, y, z = 0, \alpha = \alpha_{\text{MF, odd}}). \tag{12c}$$

The $\mathbb{Z}_2$ symmetry of the cavity-BEC system corresponds to two energetically degenerate states, which are distinguishable by a $\pi$ phase shift of the intracavity field $\alpha_{\text{MF, even}} = -\alpha_{\text{MF, odd}}$. Here we explicitly choose the one corresponding to the odd configuration, whose lattice sites are located at the desired positions $(0, \pm\lambda_c/2)$ and $(\pm\lambda_c/2, 0)$. The four-well potential is compared to the original lattice in Fig. 4. Indeed, the shape of each of the four wells precisely recreates the shape of the each lattice site of the original optical lattice.

With four sites in total and each containing $\nu \approx 25$ atoms, we perform the simulations with $\tilde{N} = 100$ atoms and $\tilde{M} = 4$ orbitals subject to the one-body potential $V_{\text{4well}}$ and contact interaction with strength $g_{\text{2D}} = 0.34\hbar^2/m_{\text{Rb}}$. MCTDH-X generates a numerically highly accurate many-body wave function for the four-well system. In terms of the quantities related to the SSF–SMI transition, for example the momentum space density distribution and the one-body correlation function between neighboring sites, the four-well model Eq. (12) should produce the same result as the full three-dimensional experimental setup from Eqs. (1) and (2). A summary of the simulation approaches and parameters is given in Appendix A.

The representation of the full effective optical lattice by our four-well model is a crucial, non-trivial simplification. The complexity of the minimal model is mainly determined by the symmetry of the full system and the filling factor $\nu$. The symmetry of the checkerboard lattice contributes significantly to the simplicity of the minimal representative four-well model. In contrast, for a system with weaker symmetry, the number of lattice sites in a unit cell $N_{\text{site}}$ increases. This will significantly increase the computational workload, which scales as $\binom{N_{\text{site}}(\nu-1)+1}{N_{\text{site}}}$.

Moreover, the validity of this simplification is based on the nature of the SSF-SMI phase transition and the geometry of the system. The finite size effect of this minimal representation for the full lattice system still proves to be the main source of systematic errors in the simplification scheme. More specifically, the transition point of a Bose-Hubbard model is subject to finite size effect, and is increased by tens of percents in terms of the ratio $t/U$ compared to the thermodynamic limit [57, 58]. However, in a cavity-BEC system, the ratio $t/U$ decreases exponentially as the pump strength $E_p$ increases [23]. As a result, the shift of the SSF–SMI boundary due to finite size effect in terms of $E_p$ is negligible. As a confirmation, we compare the phase boundary obtained for different numbers of lattice sites in Appendix E. The simulated boundaries show a systematic variance of roughly $0.5E_{\text{rec}}$, and the result has indeed already converged with the four-well model.

## 4 Results

### 4.1 The Mott transition

The momentum space density distribution $\tilde{\rho}(\mathbf{k})$ measured from experiments and calculated from simulations can be used to extract the SSF–SMI phase boundary. The obtained phase diagram of the cavity-BEC system against pump strength $E_p$ and effective detuning $\Delta_{\text{eff}}$ is shown in Fig. 2. It serves as a map to identify the three different phases of matter, NP, SSF, and SMI, which are realized in both experiments and simulations. To illustrate the system behavior in the three different phases, we choose a series of states at $\Delta_{\text{eff}} = -2\pi \times 30$ kHz, and show their simulated and experimentally measured density distributions in Fig. 5. The numbering (1 to 6) of the quantum states in Fig. 5 refers to the different pump strengths indicated in Fig. 6(b).

In the normal phase, the real space density distribution of the BEC has a Thomas-Fermi profile [Fig. 5(m)], whereas the momentum space density distribution has correspondingly a single blob [Fig. 5(a,g)]. This can be observed both in experiments and simulations. The momentum space distribution has an elliptical shape in experiments but a circular shape in simulation. This is because the harmonic trap is anisotropic in the experimental setup $\omega_x \neq \omega_y$, while the confining potential in simulations [Eq. (12b)] is isotropic in the $x$ and $y$ directions.

As the pump and hence the cavity-atom coupling are switched on, the momentum space density distributions $\tilde{\rho}(\mathbf{k})$ behave similarly in experiments [Fig. 5(a-f)] and simulations Fig. 5(g-l)], except for the thermal background in experiments which is due to heating of the sample and cannot be captured by our model. In both experiments and simulations, $\tilde{\rho}(\mathbf{k})$ in the SSF and SMI phases are completely different. It provides a way to determine the phase boundary. A typical SSF state is represented by "State 2", whose measured and simulated momentum space densities are shown in Figs. 5(b) and 5(h), respectively. In the SSF phase, the central Bragg peak at $(k_x, k_y) = (0, 0)$ is high and narrow and the satellite peaks are clearly visible. The four Bragg peaks at $(k_x, k_y) = (\pm k_c, \pm k_c)$ are the next dominant peaks and they indicate a strong coherence between atoms in the immediately neighboring sites of the checkerboard lattice, which are $(\pm \lambda_c/2, \pm \lambda_c/2)$ apart from each other in the real space. On top of these strong

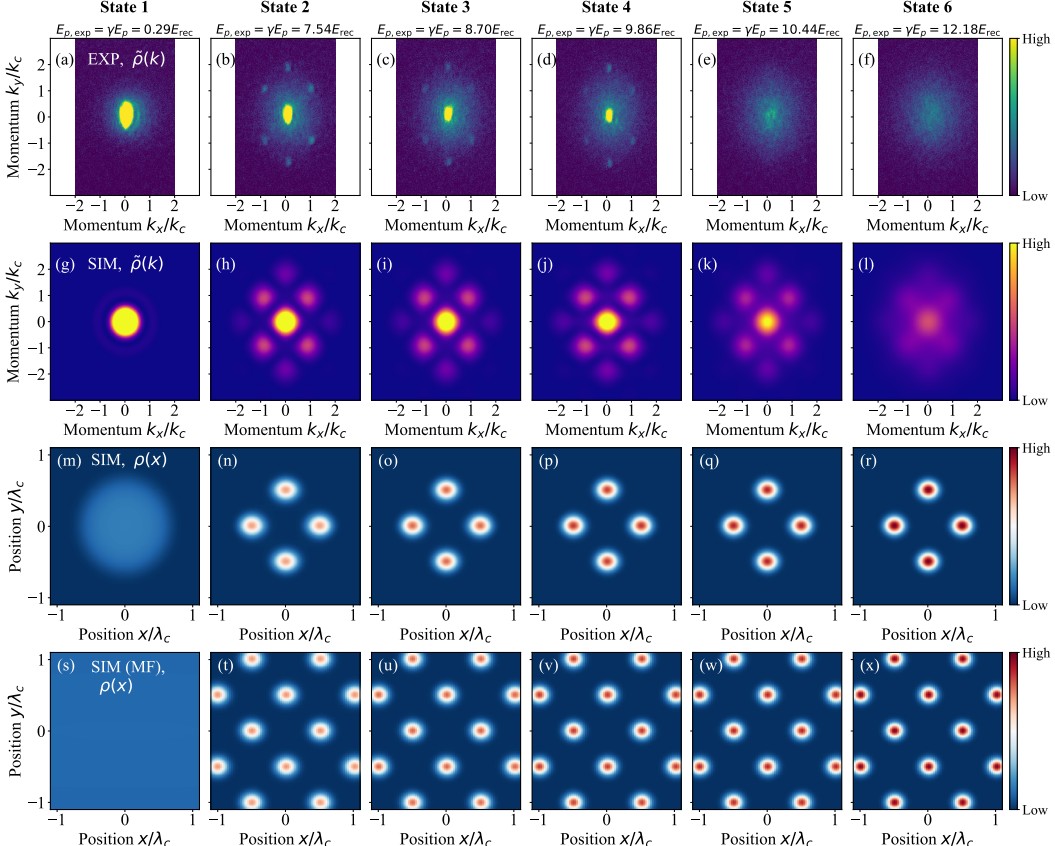

Figure 5: (a-f) Experimentally measured and (g-l) simulated momentum space density distributions $\tilde{\rho}(k_x, k_y)$, as well as simulated real space density distributions $\rho(x, y)$ of the *four-well model* for six different parameter sets. The chosen states range from normal BEC state to SSF states to SMI states. (s-x) For comparison, $\rho(x, y)$ of the *full two-dimensional model* [cf. Eq. (9)] simulated in the mean-field limit. To facilitate the comparison with panels (m-r), we zoom around the trap center and choose only the odd configurations. These states are simulated or measured at $\Delta_{\text{eff}} = -2\pi \times 30$ kHz and at different pump strengths $E_p$, which correspond respectively to the points 1 to 6 indicated in Fig. 6(b). In panel (m) the BEC is highly localized at the trap center due to the tight trap [Eq. (12b)], whereas in panel (s), the BEC has an almost uniform non-zero distribution in the center of the trap due to the relatively loose harmonic trap.

peaks, small peaks are seen at $(k_x, k_y) = (0, \pm 2k_c)$ and barely visible at $(k_x, k_y) = (\pm 2k_c, 0)$, which correspond to the optical pump lattice and the intracavity lattice, respectively. In contrast, "State 6" is a good representative of the SMI phase, whose measured and simulated momentum space densities are shown in Figs. 5(f) and 5(l), respectively. In the SMI phase, the central peak becomes broad and low, and the satellite peaks become diffuse. They indicate the strong localization of the atoms in the individual lattice sites and the lack of coherence between the atoms [15–18, 22–24, 59–63]. The localization and loss of coherence of the atoms accompanying the increasing pump strength does not trigger qualitative changes in the real space density distributions $\rho(\mathbf{x})$ [Fig. 5(m-r)], despite their significant impact in momentum space. These images are not available from our experimental setup because the resolution of the absorption imaging system in the experiment is not good enough to resolve the individual lattice sites. Nevertheless, as a sanity check, we confirm that the four-well model and

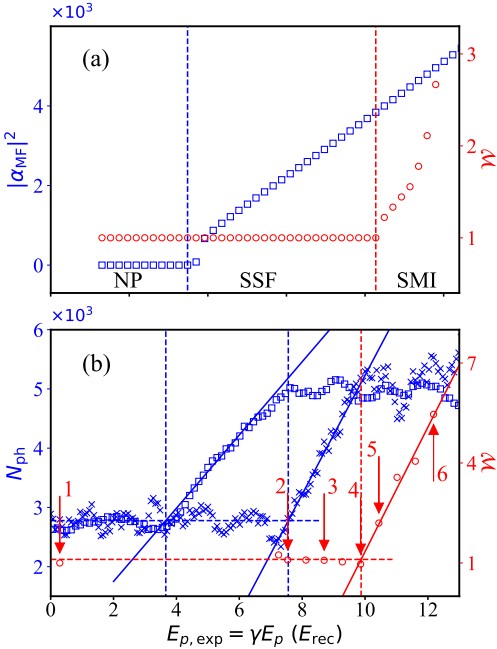

Figure 6: Cavity field magnitude $|\alpha_{MF}|^2$, intracavity photon number $N_{ph}$, and the relative width of the central Bragg peak $\mathcal{W}$ as functions of pump strength $E_{p,exp} = \gamma E_p$ at fixed detuning $\Delta_{eff} = -2\pi \times 30$ kHz. The relative width of the central Bragg peak for the BEC is set to be $\mathcal{W} = 1$. As the pump strength increases, the system transitions through all three phases, i.e., NP, SSF, and SMI. Panel (a) shows simulated steady-state results for $|\alpha_{MF}|^2$ (blue squares) obtained from the mean-field $M = 1$ simulations, and $\mathcal{W}$ (red circles) obtained from the beyond-mean-field $M = 4$ simulations. Panel (b) shows experimentally measured $N_{ph}$ for the $T_r = 40$ ms ramping protocol (blue squares), as well as $N_{ph}$ (blue crosses) and $\mathcal{W}$ (red circles) for the $T_r = 20$ ms ramping protocol. A background radiation of $N_{ph} \approx 2.7 \times 10^3$ originates from the diffuse light forming the external lattice potential. This background count offset can be safely subtracted for the determination of the NP–SSF boundary. The numbers indicate the representative points whose simulated and measured density distributions are shown in Fig. 5.

the full two-dimensional mean-field model [Fig. 5(s-x)] produce the same real space density distributions locally at the center of the harmonic trap.

For different pump strengths at the fixed detuning $\Delta_{eff} = -2\pi \times 30$ kHz, we summarize the simulated cavity field magnitude $|\alpha_{MF}|^2$, the measured intracavity photon number $N_{ph}$, as well as the simulated and measured relative widths $\mathcal{W}$ of the central Bragg peak in Fig. 6. In both simulations [Fig. 6(a)] and experiments [Fig. 6(b)], the NP–SSF boundary is defined as where $|\alpha_{MF}|^2$ or $N_{ph}$ starts to increase, whereas the SSF–SMI boundary is defined as where $\mathcal{W}$ starts to increase. Specifically for the experimental SSF–SMI boundary, we fit a line, shown as the red solid line in Fig. 6(b), using the first five data points after $\mathcal{W}$ starts to increase. The crossing with the initial width, i.e., the horizontal red dashed line, marks the SSF to SMI phase boundary.

We further discuss the discrepancies between the experimental and simulation results. The retardation effect discussed in Sec. 2.2 and Appendix B clearly manifests itself in $N_{ph}$ for the fast ramp. This dynamical effect, however, cannot be reproduced numerically because it requires a prohibitively large amount of computational resources. To better appreciate the retardation effect, we perform a fit on the respective $N_{ph}$ for both ramps, shown as blue solid

lines in Fig. 6(b). These two fitted lines have different slopes, and when they intersect, the retarded cavity field is expected to reach its steady-state value. Compared to the simulation results, a plateau is further seen in $N_{ph}$ for large pump strengths in experiments [Fig. 6(b)] for both fast and slow ramps as a result of atom loss. For the fast ramp, this plateau occurs slightly later than the increase in $\mathcal{W}$, and it thus contributes negligibly to the position of the SSF–SMI boundary. However, it contributes to one of the two factors why the increase behavior of $\mathcal{W}$ is also different between experiments and simulations in the SMI phase. The other factor is the different shapes of the initial BEC cloud in experiments and simulations due to the different confining potentials. The above analysis for $\Delta_{eff} = -2\pi \times 30$ kHz helps us comprehend the trade-off we encountered when choosing the ramping time. Even with a fast ramp $T_r = 20$ ms, the estimated position of the SSF–SMI boundary still suffers non-negligible systematic errors from the retardation effect. However, the heating and atom loss are anticipated to immediately set in and introduce further systematic errors for a slightly slower ramp.

The experimental and simulated SSF–SMI boundaries for all detunings are obtained in a similar manner, and shown in Fig. 2 as black and blue lines, respectively. Here we briefly discuss the effects of the retardation and atom loss for different detunings. The retardation effects are dominating for less negative detunings and are secondary for more negative detunings. In particular, for the least negative detunings, the steady-state SSF–SMI boundary could take place earlier than the dynamical NP–SSF boundary for the fast ramp. On the contrary, the atom loss has a larger impact for more negative detunings, where a larger pump strength is required for the self-organization and Mott insulation. We thus generally expect that the experimentally measured SSF–SMI boundary takes place at a slightly larger pump strength than the steady-state boundary. With all the experimental and simulation systematic errors under consideration, our results show that the experimental and simulation results are generally in good agreement for more negative detunings.

### 4.2   Comparison to Wannier-based Bose-Hubbard approaches

The quantitative determination of the SSF–SMI boundary can also be achieved by other existing approaches besides our MCTDH-X-based approach proposed above. We describe here an alternative approach which is based on the mapping to the Bose-Hubbard model. Given the effective optical lattice potential, the Wannier functions can be estimated by different numerical methods, many of which are available for quantum optical systems [78–81]. The Wannier functions then allow the extraction of the Bose-Hubbard parameters $t$ and $U$ [cf. Eq. (4)], which can be further used to determine the superfluid–Mott-insulator boundary. The last step can be performed by utilizing an empirical formula [65]. For the prediction of the Mott boundary, the criterion based on the Bose-Hubbard ratio $U/t$ has been shown to be compatible with the criterion based on the behaviors of the central Bragg peak [59]. This approach no longer suffers from the finite size effect in comparison to our proposed simplification scheme. Nevertheless, when calculating the Wannier functions, the broadening of Wannier function induced by on-site interaction is generally not taken into account [82, 83]. This could result in an underestimation in the Bose-Hubbard ratio $t/U$, and give rise to a different kind of systematic errors in comparison to our proposed MCTDH-X scheme.

In Appendix F, we compare the SSF–SMI boundaries obtained using the maximally localized generalized Wannier states package [80] and the MCTDH-X approach for different values of contact interaction strengths $g_{2D}$. Thereby, we validate the MCTDH-X approach at weak contact interaction. More importantly, we confirm that MCTDH-X can indeed capture higher order corrections in comparison to the Wannier-based Bose-Hubbard approach, which mainly include the expansion of the local atomic cloud at lattice sites induced by the experimentally relevant contact interaction. Eventually, these corrections have noticeable impact on the predicted position of the SSF–SMI boundary.

## 5 Conclusions

We have used MCTDH-X to quantitatively determine the SSF–SMI boundary of a recoil resolved cavity-BEC system. This is the first time that MCTDH-X simulation results are directly compared quantitatively to experimental results for a cavity-BEC system, and the comparison is non-trivial due to limitation in computational resources. In contrast to the significant dynamical effects at play and a relatively large size of the lattice in the experiments, our two-dimensional simulations are limited to steady states and a small number of lattice sites. These computational difficulties can be judiciously circumvented by choosing different ramping rates for the measurement of different quantities on the experimental side, as well as simplifying the full lattice to a minimal four-well representation in the simulation. The systematic errors of our proposed approach mainly stem from the small size of the lattice system used in simulations, and are small when expressed in terms of the pump rate. We have thereby established MCTDH-X as a feasible numerical method for the quantitative calculation of the superfluid–Mott-insulator boundary in an ultracold atomic system which forms a lattice with a large number of atoms per site.

## Acknowledgements

We acknowledge the computation time on the ETH Euler cluster and the Hawk cluster at the HLRS Stuttgart.

**Funding information** R.L. and R.Ch. acknowledge funding from the Swiss National Science Foundation (SNSF) and the ETH Grants; Ch.G., J.K. and A.H. acknowledge funding from the Deutsche Forschungsgemeinschaft (DFG, German Research Foundation) under grant SFB 925, and H.K. acknowledges DFG for funding through grant DFGKE2481/1-1; P.M. acknowledges funding from the ESPRC Grant no. EP/P009565/1; M.B. and A.U.J.L. acknowledge funding from the Austrian Science Foundation (FWF) under grant P32033.

## A  Summary of methods and parameters

The methods and parameters used in the experiments and simulations are summarized in Table 1.

## B  Experimental phase diagram with fast ramping protocol

For a complementary comparison between the slow and fast ramping protocols in experiments, we show in Fig. 7 the phase diagram for the fast ramping protocol with $T_r = 20$ ms. Compared to the steady-state phase diagram shown in Fig. 2 of the main text, the dynamical NP–SSF boundary for the fast protocol is indeed apparently shifted to higher pump strength due to retardation effect during the self-organization process. Importantly, at less negative detunings $\Delta_{\text{eff}} = -12.5$ kHz and $-17.5$ kHz, the onset of the self-organization in experiments takes place later than the loss of superfluidity predicted by the simulations. As discussed in the main text, this accounts for the discrepancy between the simulated and experimental SSF–SMI boundaries.

Table 1: Summary of the experimental and computational methods and parameters. Here $a_B$ is the Bohr radius.

| | Experiments | Simulation Step 1 "Full two-dimensional model" | Simulation Step 2 "Four-well model" |
|---|---|---|---|
| **Methods** | | | |
| Hamiltonians and/or Equations of motion | Eqs. (1), (2) | Eqs. (1), (2) with $z = 0$ | Eq. (11) |
| Solved state | Evolving state | Steady state | Ground state |
| Extracted quantities | $N_{\text{ph}}$, $\tilde{\rho}(\mathbf{k})$ | $\alpha_{\text{MF}}$ | $\tilde{\rho}(\mathbf{k})$, $\rho(\mathbf{x})$ |
| **General setup parameters** | | | |
| Particle number | $N_{\text{3D}} = 5.5 \times 10^4$ | $N_{\text{2D}} = 6{,}900$ | $\tilde{N} = 100$ |
| Orbital number | – | $M = 1$ | $\tilde{M} = 4$ |
| Particle mass | $m_{\text{Rb}} = 1.44 \times 10^{-25}$ kg | | |
| Dimensions | 3 | 2 | 2 |
| Interaction strength | $g \approx 98 \times 4\pi\hbar^2 a_B/m_{\text{Rb}}$ [84] | $g_{\text{2D}} = 0.34\hbar^2/m_{\text{Rb}}$ | $g_{\text{2D}} = 0.34\hbar^2/m_{\text{Rb}}$ |
| Trap | Harmonic with $(\omega_x, \omega_y, \omega_z) =$ $2\pi \times (25.2, 202.2, 215.6)$ Hz | Harmonic with $(\omega_x, \omega_y) = 2\pi \times (25.2, 202.2)$ Hz | Octic with form Eq. (12b) |
| **Parameters related to optical lattice potentials** | | | |
| Wave length | $\lambda_c = 803$ nm | | |
| Recoil energy | $E_{\text{rec}} = \hbar \times 2\pi \times 3.55$ kHz | | |
| Pump strength | $E_{p,\text{exp}} = 0$ to $14.5E_{\text{rec}}$ | $E_p = 0$ to $11E_{\text{rec}}$ | – |
| Single photon light shift | $U_0 = -2\pi \times 0.36$ Hz | $\tilde{U}_0 = -2\pi \times 2.87$ Hz | – |
| Effective detunings | $\Delta_{\text{eff}} = 0$ to $-2\pi \times 40$ kHz | $\Delta_{\text{eff}} = 0$ to $-2\pi \times 40$ kHz | – |
| Cavity decay rate | $\kappa = 2\pi \times 4.45$ kHz | $\kappa = 2\pi \times 4.45$ kHz | – |
| Other input parameters | – | – | $\alpha_{\text{MF}}$ obtained from Step 1 |

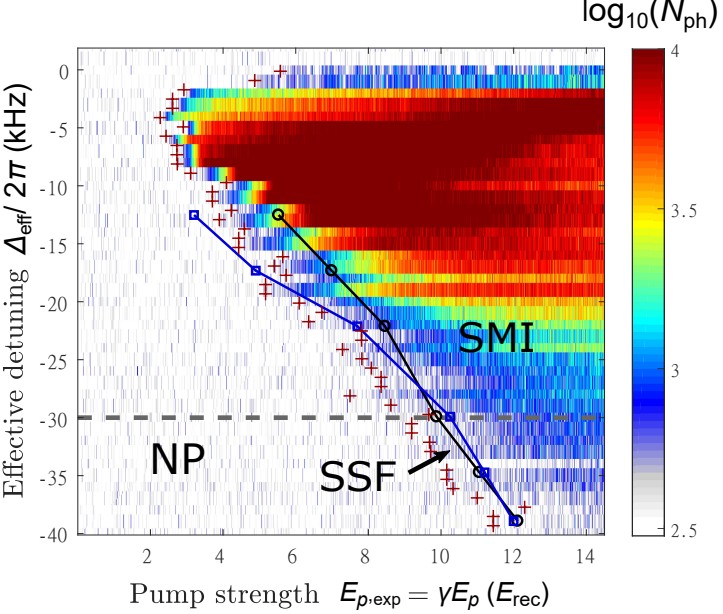

Figure 7: Experimental phase diagram for the fast ramping protocol with ramping time $T_r = 20$ ms. The photon number is shown as color scale in the background, and the NP–SSF boundary is shown as dark red crosses. The SSF–SMI boundary is shown as black circles. For comparison, the simulated SSF–SMI boundary is shown by the blue squares. The experimental and simulation data for the SSF–SMI boundary are also used for the steady-state phase diagram in Fig. 2 of the main text.

## C  Multiconfigurational time-dependent Hartree method for indistinguishable particles

The Multiconfigurational Time-Dependent Hartree Method for Indistinguishable Particles [38–40, 42, 43] is implemented in the MCTDH-X software [41], and can accurately simulate cavity-BEC systems. We consider a general Hamiltonian containing a one-body potential $V(\mathbf{x})$ and a two-body interaction $W(\mathbf{x}, \mathbf{x}')$:

$$\hat{\mathcal{H}} = \int d\mathbf{x}\hat{\Psi}^\dagger(\mathbf{x})\left\{\frac{p^2}{2m} + V(\mathbf{x})\right\}\hat{\Psi}(\mathbf{x}) + \frac{1}{2}\int d\mathbf{x}\,d\mathbf{x}'\,\hat{\Psi}^\dagger(\mathbf{x})\hat{\Psi}^\dagger(\mathbf{x}')W(\mathbf{x}, \mathbf{x}')\hat{\Psi}(\mathbf{x})\hat{\Psi}(\mathbf{x}'). \quad (13)$$

With the MCTDH-X approach, the many-body wave function follows the ansatz

$$|\Psi(t)\rangle = \sum_{\mathbf{n}} C_{\mathbf{n}}(t)\prod_{k=1}^{M}\left[\frac{\left(\hat{b}_k^\dagger(t)\right)^{n_k}}{\sqrt{n_k!}}\right]|\text{vac}\rangle, \quad (14)$$

where $|\text{vac}\rangle$ is the vacuum state, $M$ is the number of single-particle wave functions (orbitals) and $\mathbf{n} = (n_1, n_2, ..., n_M)$ gives the number of atoms in each orbital. Their sum is the total number of particles in the system $\sum_{k=1}^{M} n_k = N$. The time-dependent operator $\hat{b}_k^\dagger(k)$ creates one atom in the $k$-th orbital $\psi_k(x)$

$$\hat{b}_k^\dagger(t) = \int \psi_k^*(\mathbf{x}; t)\hat{\Psi}^\dagger(\mathbf{x}; t)dx. \quad (15)$$

The MCDTH-X working differential equations governing the time evolution of the coefficients $C_{\mathbf{n}}(t)$ and the orbitals $\psi_k(\mathbf{x}; t)$ are obtained using the time-dependent variational principle [85].

## D  Calculation of the effective two-dimensional atomic contact interaction strength

The atomic contact interaction strength in the two-dimensional system is calculated based on the Thomas-Fermi approximation. By assuming a strong interaction and comparatively vanishing kinetic energy, the Gross-Pitaevskii equation for the Thomas-Fermi cloud can be written as

$$E_0\phi(x, y) = V_{\text{trap}}(x, y)\phi(x, y) + Ng_{2D}|\phi(x, y)|^2\phi(x, y), \quad (16)$$

where $\phi(x, y)$ is the single-particle wave function, $E_0$ is the energy of the system to be determined, $N$ is the atom number, $V_{\text{trap}} = \frac{m}{2}(\omega_x^2 x^2 + \omega_y^2 y^2)$ is the harmonic trap, $g_{2D}$ is the effective two-dimensional interaction strength which we want to calculate. The atomic density profile $\rho(x, y) = |\phi(x, y)|^2$ thus follows:

$$\rho(x, y) \equiv |\phi(x, y)|^2 = -\frac{m}{2Ng_{2D}}(\omega_x^2 x^2 + \omega_y^2 y^2) + E_0, \quad (17)$$

and vanishes at $\rho(r_x, 0) = \rho(0, r_y) = 0$, with the Thomas-Fermi radii

$$\omega_x r_x = \omega_y r_y = \sqrt{\frac{2E_0Ng_{2D}}{m}} \equiv \tilde{r}_0. \quad (18)$$

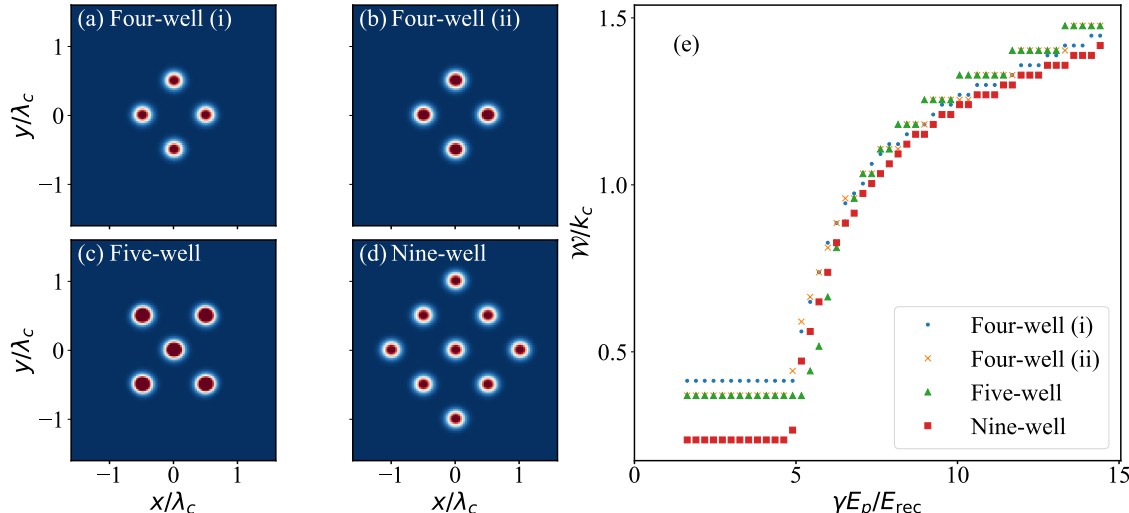

Figure 8: (a-d) Exemplary real-space density distributions of (a) the four-well system described by Eq. (12b) in Sec. 3.3, (b) an alternative realization of the four-well system, (c) a five-well system, and (d) a nine-well system. (e) The width $\mathcal{W}$ of the central peak in the momentum space as a function of the pump strength $E_p$ in these four systems.

The normalization of the density distribution requires $\int_\Omega dx\,dy\,\rho(x,y) = 1$ where the integration region $\Omega = \{x,y|\rho(x,y) > 0\} = \{x,y|\omega_x^2 r_x^2 + \omega_y^2 y^2 < \tilde{r}_0^2\}$ is an ellipse. This solves the system energy:

$$E_0 = \sqrt{\frac{m\omega_x\omega_y}{\pi N g}}\,. \tag{19}$$

Combining the equations above, the two-dimensional contact interaction strength is

$$Ng_{2\mathrm{D}} = \frac{\pi m}{4}\frac{r_x^4\omega_x^3}{\omega_y} = \frac{\pi m}{4}\frac{r_y^4\omega_y^3}{\omega_x}\,. \tag{20}$$

## E  Effects induced by the confining potential and the size of the reduced lattice

In Sec. 3.3 we have argued that, for the purpose of determining the SSF–SMI boundary in terms of $E_p$, it is enough to use the unit cell with four sites to represent the full lattice. This is because size effect only slightly affects the transition point in terms of the Bose-Hubbard parameter ratio $t/U$, which in turn has an exponential dependence on the pump strength $E_p$. Furthermore, we have argued that the simulated boundary is almost insensitive to the confining potential $V_{\mathrm{conf}}$, which we use to impose the boundary condition for the small lattice cell. In this Appendix we confirm these arguments by performing simulations with a different number of lattice sites and different confining potentials, and the results are summarized in Fig. 8.

We reproduce the simulations in Sec. 3.3 with confining potentials different from the one presented in Eq. (12b). The confining potentials we use for Fig. 8 share the form

$$V_{\mathrm{conf}}(x,y) = E_{\mathrm{conf}}(x^2 + y^2)^8/\lambda_c^{16}\,. \tag{21}$$

For a fixed effective detuning $\Delta_{\text{eff}} = -2\pi \times 30$ kHz and varying pump strength $E_p$, we use different confining potential strengths $E_{\text{conf}}$ on top of the optical lattice $\tilde{V}_{\text{opt}}(x, y)$, and choose the configuration parity of the lattice according to our need. The following combinations of confining potential strengths and lattice configuration parities are chosen:

$$E_{\text{conf,4well(ii)}} = 20E_{\text{rec}}, \qquad \text{odd lattice}, \tag{22a}$$

$$E_{\text{conf,5well}} = 10E_{\text{rec}}, \qquad \text{even lattice}, \tag{22b}$$

$$E_{\text{conf,9well}} = 0.01E_{\text{rec}}, \qquad \text{even lattice}. \tag{22c}$$

These combinations respectively produce an alternative realization of the four-well system, a five-well system, and a nine-well system. Their exemplary real-space density distributions are shown in Fig. 8(a-d).

In order to make the computational effort feasible and the simulation results comparable, we impose a filling factor of $\nu = 1$ for all the four cases. As a result, the number of atoms $N$ and the number of orbitals $M$ are both equal to the number of wells in the simulations. We summarize the width $\mathcal{W}$ of the central peak in the momentum space for different confining potentials in Fig. 8(e).

In the SSF phase, the width $\mathcal{W}$ is sensitive to the confining potential, and similar effects have been seen in Fig. 6 in the main text. This sensitivity contributes to a slight variance in the predicted SSF–SMI boundary for different confining potentials. Nevertheless, in all the four scenarios that we investigate in this Appendix, the SSF–SMI boundary is predicted to take place at roughly the same pump strength, with a variance of roughly $0.5E_{\text{rec}}$. We can thus confirm that, for the determination of the SSF–SMI boundary in terms of the pump strength, a small system with four lattice sites is enough and the sensitivity on the form of the confining potential is small. We note that now the SSF–SMI transition takes place at a smaller pump strength than the results in Fig. 6(a) because of the low filling factor.

## F Comparison to maximally localized generalized Wannier states method

We compare here the MCTDH-X predictions of the SSF–SMI boundary with the maximally localized generalized Wannier states (MLGWS) package [80] used for optical lattice potentials. We consider the optical lattice potential $\tilde{V}_{\text{opt}}$ obtained in Eq. (12c) for different pump strengths $E_p$ at fixed detuning $\Delta_{\text{eff}} = -2\pi \times 30$ kHz. Using the MLGWS method, we obtain the ratio of the Bose-Hubbard parameters $U/g_{\text{2D}}t$ in units of $m_{\text{Rb}}/\hbar^2$ as shown in Fig. 9(a). The SSF–SMI boundary of the two-dimensional system can then be determined using the empirical formula presented in Ref. [65]:

$$(U/t)_c = 2\nu(5.80 + 2.66\nu^{-2.19}). \tag{23}$$

For the experimentally appropriate filling factor of $\nu = 25$, the SSF-SMI boundary is estimated to be $(U/t)_c \approx 290$. This threshold is indicated as red squares in Fig. 9(a) for different values of $g_{\text{2D}}$. We emphasize that the MLGWS method does not take into account the broadening effects of the Wannier functions due to the finite contact interaction $g_{\text{2D}}$. Consequently, the curve $U/g_{\text{2D}}t$ remains unchanged for different values of $g_{\text{2D}}$. The method can significantly underestimate the hopping strength $t$ and thus predicts that the SSF–SMI transition takes place at a much smaller pump strength.

The MCTDH-X predictions are obtained from the width $\mathcal{W}$ of the central peak in the momentum space density for different values of contact interaction $g_{\text{2D}}$. These simulations are

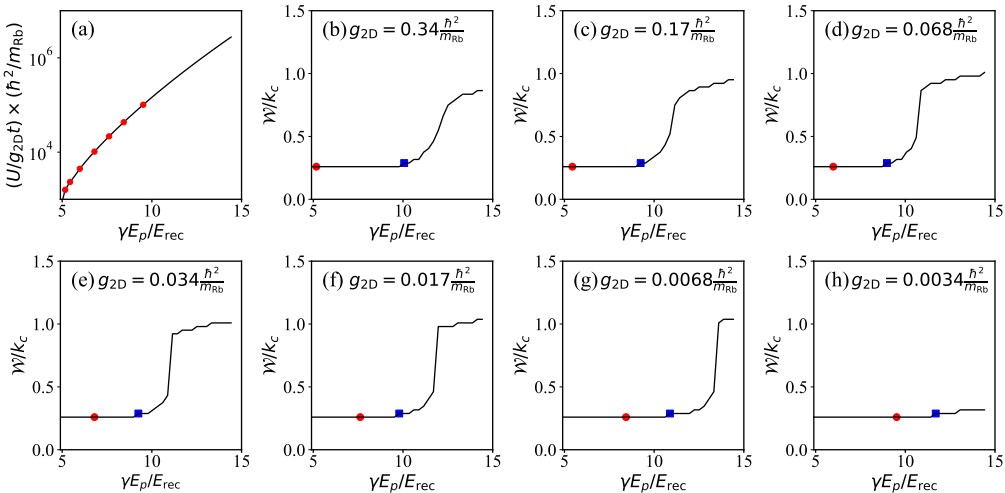

Figure 9: (a) The ratio of the Bose-Hubbard parameters $U/g_{2D}t$ as a function of the pump strength $E_p$ calculated using the MLGWS package [80]. The red squares indicate the predicted Mott transition point $(U/t)_c \approx 290$ [cf. Eq. (23)] for different values of contact interaction strength: (from left to right) $g_{2D}m_{Rb}/\hbar^2 = 0.34$, 0.17, 0.068, 0.034, 0.017, 0.0068, 0.0034. (b-h) The width $\mathcal{W}$ of the central peak in the momentum space as a function of pump strength $E_p$ for different values of contact interactions $g_{2D}$ simulated using MCTDH-X. The Mott transition is determined by the onset of the increase in $\mathcal{W}$, and is indicated in the figures by the blue squares. The Mott transition points predicted in panel (a) are also shown in the corresponding panels (b-h) as red squares.

performed at different pump strengths $E_p$ at fixed detuning $\Delta_{\text{eff}} = -2\pi \times 30$ kHz. Fig. 9(b-h) presents a comparison of the SSF–SMI boundaries obtained using both methods. For the large, experimentally relevant value $g_{2D} = 0.34\hbar^2/m_{Rb}$ [Fig. 9(b)], the MCTDH-X boundary occurs at a much larger pump strength than that predicted by MLGWS. This difference can be attributed to the realistic width of the local atomic clouds at each individual lattice site. As $g_{2D}$ decreases, the boundaries predicted by the two methods approach each other. Nonetheless, a difference of $\gamma \Delta E_p \approx 2E_{\text{rec}}$ between the two boundaries still remains at small contact interaction $g_{2D} = 0.0034\hbar^2/m_{Rb}$ [Fig. 9(h)]. A potential source of this discord is the non-trivial effects induced by the trapping potential, which are not considered by the MLGWS package.

Interestingly, the position of the boundary predicted by MCTDH-X does not move monotonically as $g_{2D}$ decreases. In particular, when $g_{2D}$ decreases from $0.34\hbar^2/m_{Rb}$ to $0.068\hbar^2/m_{Rb}$, we observe that the MCTDH-X boundary moves slightly towards shallower optical lattice potential depths, which is contradictory to straightforward expectations. This indicates that, as the contact interaction $g_{2D}$ decreases, the increase in the hopping strength $t$ due to atomic cloud expansion dominates over the decrease in the on-site interaction $U$ in this regime. This result is in consistent with the findings of Ref. [82], where $t$ has been observed to increase more significantly with $g_{2D}$ for a larger value of $\nu g_{2D}$.

We have thereby confirmed the consistency between the MLGWS package and the MCTDH-X scheme at weak contact interaction. More importantly, we have confirmed that the MCTDH-X scheme can incorporate higher order effects induced by a strong contact interaction.

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
