# Peer review of "Mott transition in a cavity-boson system: A quantitative comparison between theory and experiment"

_SciPost Physics, doi:SciPost Phys. 11, 030 (2021)_

## Round 1 · Referee Report · Anonymous (Referee 1) · 2021-5-24

Strengths
2-Introduction of new computational method, potentially applicable to many other experimatal scenarios
Weaknesses
2-A direct comparison to traditionally employed theoretical techniques is missing
Report
The authors present a comparison of theoretical and experimental phase boundaries employing MCTDH-X, reducing computational complexity by reducing the atomic system to four wells of the optical lattice.
The employed theoretical tools are novel and have the potential to be employed in many other experimental scenarios, making them quite appealing for the simplicity and at the same time effectiveness of the method.
I believe the paper is well written and deserves publication. I have several comments that would improve the quality of the manuscript.
Requested changes
1-In section 2.2, concerning retadation effects, the authors take the strategy of measuring the different phase boundaries with different ramping times. For the NP-SSF transition, they employ a slow ramp, to avoid retardation, while for the SSF-SMI they use a faster ramp to minimize atom loss. In this case, the light field suffers from retardation as reported in the methods section. The atomic potential is altered by the presence of the cavity light field and this retardation also affects the SSF-SMI experimental phase boundary. A discussion of this effect already in this section would improve the clarity of the manuscript on this point.
2-In the same section, concerning the calibration factor applied to the effective pump strength, the authors argue for it from the point of view of the pump not being monochromatic. An indication of the laser linewidth would help the reader to understand this point.
3-The authors write that "only the component at frequency $\omega_p$ can scatter into the cavity", but such frequency is defined as the frequency of the pump itself. Do the authors imply that the spectrum of the pump is being filtered? If so, how broad is the frequency range involved in the scattering process? Does the comparison of frequency range to the laser linewidth approximately justify the value the authors report for the calibration factor?
4-In section 4, caption of figure 5, there is a mention of relatively strong background signal picked up by the detector as soon as the laser is switched on, in fact making up about half of the maximum photon number recorded during self-organization. A discussion of this point is missing in the text. Does such signal come to the detector as diffused light or does it come form the cavity? In this last case, it would induce a strong symmetry breaking on the transition, possibly substantially modifying the behaviour of the system.
5-In the conclusions, the authors compare their method to a more traditional approach based on the Bose-Hubbard model. They regard it as complementary to theirs, suffering from different systematic effecs with respect to the ones their approach suffers from. I found this point very informative, in fact it would make sense in my opinion to move the discussion to section 3, or even the introduction, in order to motivate the author’s choices in this respect. It would also make sense to have a direct comparison between the two approaches, at least in the methods section, to appreciate the differences in this regard.

---

## Round 1 · Referee Report · Anonymous (Referee 2) · 2021-5-27

Strengths
1- Clear and concise presentation of the methodology and the results. 2- Comparison of numerical results obtained via a new technique with real experimental data. 3- The numerical results obtained via the MCTDH-X method agree reasonably well with the many-body quantum phases observed in the experiment. Therefore, the method could serve as a powerful tool to answer open questions in the realm of many-body cavity QED.
Weaknesses
1- The presentation of the results can be improved in certain parts (see list of required changes below). 2- Potential additional systematic errors are not elaborated in great detail.
Report
The results and methodology are presented in great detail and the focus of the main text lies on the fundamental physics and the general methodology. Details for the more expert reader are provided in a well written appendix.
Based on these arguments I think the manuscript fulfills the acceptance criteria of SciPost physics and should be published.
Nevertheless, I would like the authors to perform some minor changes on the manuscript and address the points below. This should enhance the clarity to the non-expert reader and shed some additional light on potential additional systematic errors.
Requested changes
1- Fig 5: The authors should extend the $y$-axis in Fig. 5(a) such that the values of $|\alpha|^2$ in the SMI phase are shown. The experimental data show a plateau of $N_\mathrm{ph}$ beyond a certain $E_{p,\mathrm{exp}}$. I assume the numerical $|\alpha|^2$ does not exhibit this plateau because it is obtained via a mean-field treatment. While the applied self-consistent approach is perfectly fine from a methodological point of view, it seems that the mean-field $\alpha$ leads to some systematic error when comparing the numerics to experimental data. Why do the experimental data exhibit this plateau? Could it be related to heating or particle loss (see also point 4 below)? The authors should elaborate a bit more on this point in their discussion of Fig. (5).
2- I think it would be instructive to provide similar figures as Fig. 4(g)-(l) [and maybe Fig.5(a)] for the single orbital case M = 1 (mean-field) in the appendix. This would be a powerfull display of the underlying problem which the authors address (no prediction of SMI phase) with their new method.
3- In the last paragraph of section 3.2 (page 9) the authors state that the "NP-SSF boundaries collapse upon each other" in Fig. 2. I can see this for small values of the effective detuning but for larger values the numerical and experimental curves deviate quite substantially. The same holds for Fig. 6 in the appendix. This discrepancy could be related to my arguments under point 1 in this list (see above). It would imply that the value for $|\alpha|^2$ is overestimated in the applied self-consistent method in comparison to real experimental values. This means that the numerics predicts the SSF-SMI boundary for weaker pump strenghts due to this overestimation in $|\alpha|^2$. I think the authors should comment on this point and amend the paper correspondingly.
4- The authors attribute the difference in the relative widths of the central Bragg peak $\mathcal{W}$ in the experimental data and the numerical results to the different shape of the initial BEC due to the asymmetric trapping potential in the experiment. However, one thing which is not discussed throughout the manuscipt is heating of the BEC via the pump beam in the experiment. In Fig.4(a)-(f) one can clearly see the buildup of some thermal background, which of course is not captured by the employed numerical method. Could it be that heating effects also contribute to the discrepancy in $\mathcal{W}$?
5- At several stages throughout the manuscript the authors claim that the self-organization of the BEC leads to a superlattice (see for example paragraph 2 on page 2). I don't understand why the checkerboard lattice which is formed in this case should be called a superlattice. Usually the term 'superlattice' is referred to a periodic lattice with periodically varying heights of the potential wells. I don't see where this is the case in the presented results.
6- Typos to be corrected: -abstract: quantiative -> quantitative -first line page 11: simplfication -> simplification - reference [41]: light$\hat{A}$ -> light

---

## Round 1 · Referee Report · Anonymous (Referee 3) · 2021-5-31

Strengths
2-Excellent agreement between theory and experiments.
Weaknesses
Report
Apart from minor changes, which I list below, my main criticism concerns the discussion of the proposed theoretical method. Clearly, from a theoretical point of view, the main problem in the considered system is the competition between local and global (cavity-mediated) interactions. The authors circumvent this problem by simulating a four-well model. This is the smallest lattice size where the self-ordering transition to a checkerboard phase can be observed, so it seems the authors need to know in advance which many-body phases to expect. It would be good if such limitations were discussed openly. It would be good to present an outlook how this theoretical tool will scale with different setups that lack this fundamental symmetry.
In the same spirit, the authors emphasize in the introduction that the presented MCTDH-X method "captures many-body effects beyond the Gross-Pitaevski mean field limit". What is the relevance of beyond-Gross-Pitaevskii effects? A direct comparison with mean field methods, however superficial, might be very helpful for readers to appreciate the relevance of these effects.
Requested changes
1-On page 2, the authors write "The cavity-BEC system can thus reproduce a quantum-optical version of the Bose-Hubbard model."
It's not clear what defines the "quantum-optical" version of the BH model. I suppose the authors are thinking of a global interaction akin to the cavity-dipole coupling in Jaynes-Cummings or Tavis-Cummings models? This needs to be clarified.
2-The authors write "We emphasize that once the effective contact interaction strength g2D and the calibration factor are determined, there is no more free parameter in the simulation when compared to the experimental system." How does this compare to other approaches?

---

## Round 2 · Referee Report · Anonymous (Referee 2) · 2021-7-12

Report

I would like to thank the authors for addressing my previous comments in detail and to perform related changes on the manuscript. In particular the in depth discussion of potential systematic errors, as well as, the comparison to methods based on Wannier states following the other referee's comments provide valuable additional information to the reader.

Based on this and the arguments mentioned in my previous report I recommend publication of the manuscript in the SciPost physics.

---

## Round 2 · Author Response

We would also like to thank the three Referees for their careful examination of our work. Referee 1 and 2 recommended our manuscript for publication in SciPost Physics. We have revised the manuscript and implemented their comments to improve the manuscript. Referee 3 wrote “It could there be published if the authors address my concerns”. We hope that we address in the following his/her concerns satisfactorily.

Particularly, the three Referees have pointed out two main weaknesses of our manuscript: the systematic errors from the experiments are not discussed in a detailed manner; and a more substantial comparison between our numerical methods with existing ones is missing. These two points are both more properly addressed in the new version.

In the new manuscript, the major changes related to the Referee’s comments are marked in blue. Please note that Figs. 4 and 5 in the previous version now become Figs. 5 and 6, respectively.

Reply to comments from Referee 1.

1. (Retardation effects) Indeed the retardation effect of the cavity field would also affect the SSF-SMI boundary. We now discuss it in more detail in Sec. 4.1. As we further discuss in the same section, this retardation effect is inevitable due to the dilemma between the retardation effect and the atom loss.

2 & 3. (Monochromaticity) In experiments, the main loss of the laser pump strength comes from the two side peaks due to servo electronics, which are in the order of MHz away from the central peak. This shift is much larger than the cavity dissipation rate, and thus these two side peaks cannot contribute to the scattering of the cavity field. On the other hand, the linewidth of the central peak cannot be accurately measured because we do not have another stable laser to beat with. However, we estimate that the linewidth is roughly 100Hz, which is much smaller than the cavity dissipation rate. It can thus fully scatter into the cavity. The calibration factor 1.36 seems reasonable to us. We add a few lines discussing this in Sec. 2.2.

4. (Background signal of the cavity field). This large background counts originates from diffuse scattering on the optics implemented in the system, and not from the cavity. It can thus be safely subtracted from the data for the determination of the phase boundary. We add a sentence in the caption of Fig. 6 discussing that. Moreover, as one can see in our recent preprint Ref. 54, the symmetry in our system is very well established and there is no significant breaking of the symmetry due to diffusively scattered light. We clarify this in Sec. 2.1.

5. (Comparison to existing approaches). We thank the Referee for his/her appreciation on this discussion. Inspired by this comment, we now further perform a more quantitative comparison between the MCTDH-X approach and the existing approach based on Wannier function analysis. This is now discussed in the new Appendix F. In the main text, we also move the relevant paragraph to Sec. 4.2. We decide to keep the discussion after the main result instead of moving it to an earlier section, because it requires understanding of the SSF-SMI transition, which is mainly discussed in Sec. 4.1.

Reply to comments from Referee 2.

1, 3 & 4. (Heating and atom loss). Indeed there are multiple effects induced by heating and atom loss which we have not fully discussed in the previous version.

-- (Comment 3): The NP-SSF boundary is slightly shifted to higher pump strengths for the large detunings due to heating and atom loss because we reach the self-ordered phase at larger pump strength and hence later times. Therefore, when we are fitting this boundary, we only consider the data points from low detunings (-2pi5 kHz to -2pi20 kHz). This has already been implemented for the fitting in the last version, and is now explicitly clarified in Sec. 3.2.

-- (Comments 1 & 3): Moreover, the heating and atom loss are also accountable for the plateau seen in $|\alpha|^2$ in Fig. 6(b). This will certainly affect the cavity-induced optical potential. However, for the fast ramp presented in Fig. 6(b), this plateau occurs later than the Mott transition, so it has little effect on the predicted boundary. This is discussed now in Sec. 4.1.

-- (Comment 4): However, inside the SMI phase, indeed the atom loss will change the shape of the central Bragg peak. This indeed partly contributes to the difference between the behavior of W in experiments and simulations. We now also discuss this in Sec. 4.1.

2. (Mean-field density distributions in momentum space). In the mean-field limit, if we blindly use the algorithm to calculate the “momentum-space density distribution”, we would obtain the Fourier transform of the real-space density distribution. Nevertheless, we would say that this result does not have physical significance, because it has not converged numerically. Therefore, we don’t think that it would be useful to present them.

5 & 6. (Terminology and typos) We thank the Referee for pointing out these issues. We have now replaced “superlattice” with “lattice”, and fixed the typos.

Reply to comments from Referee 3.

In the report, the Referee posts two questions regarding to the applicability of the four-well model and the relevance of the beyond-Gross-Pitaevskii mean-field effects, respectively.

1. (Applicability of the four-well model). Indeed, the symmetry of the checkerboard lattice makes us possible to choose a simple model with only four lattice sites, and this is not guaranteed for many other systems. As the number of lattice sites in a unit cell increases, the required computational resources scale up significantly. We now include an extra paragraph in Sec. 3.3 to discuss this.

2. (Beyond mean-field effects). If we constrain ourselves only in the Gross-Pitaevskii mean-field limit, we are able to reproduce the self-organization, but we cannot capture the Mott insulation. This is because the Mott insulation requires at least one single-particle orbital for each lattice site. If we blindly do the simulations in the mean-field limit even in the four-well model, we will consistently obtain “superfluid” states even for very large lattice depth. This has already been discussed in detail in Sec. 3.1, but is now also briefly discussed in the introduction as suggested by the Referee.

The Referee also requests us to clarify two statements.

1. (Quantum-optical Bose-Hubbard model). Here what we have in mind is a Bose-Hubbard model with long-range interaction between lattice sites. We have made the statement more clear now.

2. (Parameters). Indeed our sentence was a bit confusing in the previous version. We want to say that these are the last two parameters which we need to determine for the simulation. We have clarified this now. If we want to use other approaches instead, for example the Wannier-based Bose-Hubbard approach as presented in Sec. 4.2 now, the same parameters are also required.

---

## Round 2 · List of Changes

Sec. 1: We clarify the phrase “quantum-optic version of the Bose-Hubbard model”, and briefly discuss the beyond-mean-field effects in our system.

Sec. 2.2: We further discuss in detail the retardation and heating effects on both the NP-SSF and SSF-SMI boundaries. We also clarify the source of power loss of the pump laser.

Sec. 3.2: We now include a new figure showing the real space density distribution of the full 2D model. We also clarify that the fit for the experimental NP-SSF boundary only takes the data points from small detunings into account, in order to minimize the heating effects.

Sec. 3.3: We discuss how the computational effort scales when the symmetry of the optical lattice becomes weaker.

Sec. 4.1: We now include new panels (s-x) in Fig.5 showing the real space density distribution of the full 2D model. This is for comparison with the four-well model in panels (m-r). We also discuss now the retardation and heating effects observed in Fig. 6 in more detail. We have also fixed a typo in the experimental results in Fig. 6(b). The number of photons is in the order of 10^3.

Sec. 4.2 & Appendix F: We perform a comparison of our MCTDH-X approach with an alternative approach based on Wannier functions and Bose-Hubbard model. The detail of the comparison is shown in the new Appendix F.

We also rewrite several paragraphs to improve the narrative.

---

## Editorial Decision

published